# Impact of Early Proteinuria Reduction in Glomerular Disease and Decline of Kidney Function: A Retrospective Cohort

**DOI:** 10.3390/jcm11195968

**Published:** 2022-10-10

**Authors:** Filipe Marques, Joana Reis, Iolanda Godinho, Marta Pereira, Paulo Fernandes, Sofia Jorge, José António Lopes, Joana Gameiro

**Affiliations:** 1Division of Nephrology and Renal Transplantation, Department of Medicine, Centro Hospitalar Universitário Lisboa Norte, EPE, 1649-028 Lisbon, Portugal; 2Clínica Universitária de Nefrologia, Faculdade de Medicina da Universidade de Lisboa, 1649-028 Lisbon, Portugal

**Keywords:** proteinuria, glomerular disease, chronic kidney disease

## Abstract

Background: In glomerular disease, the degree of proteinuria is closely related to the progression of chronic kidney disease, and its reduction is associated with a slower decline in the glomerular filtration rate (eGFR) and consequent improvement in the renal prognosis. The aim of this study was to evaluate the impact of proteinuria reduction on the decline of the eGFR in patients with glomerular disease, during the first year after the diagnosis. Methods: This was a retrospective analysis of patients with primary glomerular disease, followed at the Nephrology Department of Centro Hospitalar Universitário Lisboa Norte, during 2019. We analyzed demographic, clinical and laboratorial characteristics (creatinine, GFR, urine analysis and quantification of proteinuria determined by the proteinuria/creatinuria ratio, in the first morning urine or a 24 h urine sample). The outcome assessed was the decline in renal function, defined as a reduction in the GFR ≥ 25%, during the follow-up period. Results: We analyzed 197 patients with glomerular disease, with a mean age of 41.7 ± 19.7 years and follow-up time of 6.5 ± 5.3 years. At the time of the diagnosis, the eGFR was 81.5 ± 49.8 mL/min/1.73 m^2^ and proteinuria was 3.5 g/24 h (IQR 5.8). At one-year follow-up, median proteinuria was 0.9 g/24 h (IQR 2.4). At the end of the follow-up, mean eGFR was 72.1 ± 43.3 mL/min/1.73 m^2^. Proteinuria (*p* = 0.435) and the eGFR (*p* = 0.880) at the time of diagnosis did not correlate with long-term decline in the eGFR. Proteinuria < 1 g/24 h (HR 0.45 (95% CI 0.25–0.83) *p* = 0.011) after the first year was protective against long-term decline in the eGFR. It maintained this association with the long-term eGFR decline, independently of the duration of the follow-up (HR 0.30 (95% CI 0.17–0.52) *p* < 0.001). Conclusions: Proteinuria reduction to lower than 1 g/24 h, during the first year after diagnosis, was a protective factor for the long-term decline of kidney function, having a more important role than proteinuria or the GFR at the time of the diagnosis.

## 1. Background

Chronic kidney disease (CKD) is a common disease, with an estimated prevalence of 8 to 16% worldwide [1], and its progression is associated with poor outcomes such as increased cardiovascular morbidity, poorer quality of life and reduced survival [2,3]. In the adult population, diabetes and arterial hypertension account for more than half of the cases of CKD [4]; however, in young adults, glomerular diseases are the main cause of CKD, mainly in male patients [5]. Glomerular diseases are the third leading cause of end-stage renal disease in Europe and United States, accounting for approximately 10,000 cases of end-stage renal disease annually [6,7].

Membranous nephropathy, focal segmental glomerulosclerosis and IgA nephropathy are the primary glomerular diseases associated with greater decline in renal function [8].

The decline in renal function is intimately related to the degree of proteinuria, the presence of arterial hypertension and the glomerular filtration rate (GFR) at the time of the diagnosis. The proteins that are filtrated by the glomerular capillaries exert direct toxic damage to the tubulointerstitial structures, by activating several genes responsible for codifying vasoactive and proinflammatory molecules, which, associated with other risk factors such as hypertension, accelerate the decline of the kidney function [9].

Therefore, proteinuria is a major risk factor for an accelerated decline of the GFR. Ruggenenti et al. studied 352 patients with nondiabetic renal disease and concluded that patients with proteinuria superior to 3.9 g/24 h at the time of the diagnosis had an association with a greater decline of the GFR, compared with patients with proteinuria lower than 1.9 g/24 h, and more frequently reached end-stage kidney disease (ESKD) [10]. Another study showed that the risk of progression to ESKD progressively increased with increasing proteinuria, becoming significant for a proteinuria concentration of 0.5–1 g/24 h (HR 1.8 and 1.85 in diabetic and nondiabetic patients, respectively) and > 1 g (HR 2.70 and 2.69 in diabetic and nondiabetic patients, respectively) [11]. A recent meta-analysis that included 28 studies showed that a reduction in proteinuria of 30%, during a 2-year follow-up period, was associated with a 22% reduction in the risk of developing ESKD [12]. Therefore, the reduction in proteinuria in patients with glomerular disease was associated with a better renal prognosis. Proteinuria is also associated with a greater risk for coronary arterial disease, cerebrovascular disease, gastrointestinal bleeding, hypercoagulability, thromboembolic events and mortality [13].

The aim of this single center study was to evaluate the impact of proteinuria reduction on the decline of the GFR in patients with glomerular disease, during the first year after the diagnosis.

## 2. Materials and Methods

This was a retrospective analysis that included adult patients with glomerular diseases, followed at the Nephrology and Renal Transplantation Department of Centro Hospitalar Universitário Lisboa Norte (CHULN) during 2019. The study included patients that were diagnosed from 2005 to 2017. This study was approved by the Ethical Committee in agreement with institutional guidelines. Due to the retrospective and noninterventional nature of the study, informed consent was waived by the Ethical Committee.

### 2.1. Participants

Eligible patients were selected as adult patients (≥18 years of age) with a diagnosis of primary glomerular disease, confirmed by kidney biopsy, with at least 2 years of follow-up.

The included glomerular diseases were minimal change disease, focal segmental glomerulosclerosis, membranous nephropathy, IgA nephropathy and membranoproliferative glomerulonephritis.

Exclusion criteria comprised patients with an estimated glomerular filtration rate (eGFR) < 15 mL/min/1.73 m^2^ at the time of clinical presentation and patients lost in the follow-up. We excluded patients with pauci-immune glomerulonephritis, neoplasia and other secondary causes of glomerulonephritis.

### 2.2. Variables and Outcomes

Patient variables were collected from individual clinical records. We analyzed several clinical variables including patient demographic characteristics (age, gender, ethnicity), laboratory values at the time of the diagnosis (serum creatinine (SCr), 24 h proteinuria measurement or urinary protein to creatine ratio in first morning urine), the histopathology result of the kidney biopsy and immunosuppressive regiment used.

The glomerular filtration rate was estimated based on the CKD-EPI formula [13]. We evaluated the decline of the eGFR during the follow-up.

Complete remission was defined as proteinuria < 0.3 g/24 h, normal SCr and serum albumin > 3.5 g/dL. Partial remission was defined as proteinuria between 0.3 g/24 h and 3.5 g/24 h, with a decline of at least 50% from the highest measurement, and stable SCr (variation < 25%).

CKD was defined as an eGFR < 60 mL/min/1.73 m^2^, persistent for at least 3 months [14].

The primary outcome was the decline of kidney function, defined as a reduction in the eGFR ≥ 25% during the follow-up. We also analyzed the reduction in the eGFR ≥ 50%, the variation of the eGFR during the follow-up and the annual decline of the eGFR.

### 2.3. Statistical Methods

Normal distribution of the variables was evaluated with a Kolmogorov–Smirnov test. Categorical variables were described as the total number and percentage for each category, whereas continuous variables were described as the mean ± standard deviation. Continuous variables were compared with the Student’s *t*-test and categorical variables were compared with a chi-square test. Variables that did not follow a normal distribution were compared using the Mann-Whitney test and described as the median and interquartile range (IQR). To calculate the probability of the decline of the eGFR dependent on proteinuria reduction, we used a Kaplan–Meier analysis as a log-rank.

We used the Cox logistic regression method to determine the variables with a significant statistical difference for the decline of the eGFR. Data were expressed as hazard ratios (HRs) with 95% confidence intervals (CIs). Statistical significance was defined as a *p*-value < 0.05. Statistical analysis was performed with the statistical software package, SPSS for Windows (version 21.0, SPSS Inc., Chicago, IL, USA).

## 3. Results

We identified 197 patients with glomerular disease. One hundred and nine (55.3%) were male, with a mean age of 41.7 ± 19.7 years (Table 1). Nearly one third of patients (31.9%, n = 63) were diagnosed with focal segmental glomerulosclerosis (FSGS), 28.4% (n = 56) with IgA nephropathy, 18.3% (n = 36) with minimal change disease, 10.7% (n = 21) with membranoproliferative glomerulonephritis and 10.7% (n = 21) with membranous nephropathy (MN) (Table 2).

At the time of the diagnosis, the mean eGFR was 81.5 ± 49.8 mL/min/1.73 m^2^, with 41.6% (n = 82) of patients having an eGFR < 60 mL/min/1.73 m^2^. Median proteinuria was 3.5 g/g or g/24 h (IQR 5.8), with 21.3% (n = 42) having proteinuria < 1 g/24 h, 24.4% (n = 54) with proteinuria of 1–3 g/24 h, 19.3% (n = 38) with 3–5 g/24 h and 35.0% (n = 69) having proteinuria ≥ 5 g/24 h (Figure 1). Patients with FSGS had a lower eGFR (52.3 ± 20.3 mL/min/1.73 m^2^) and patients with MN had higher baseline proteinuria (6.1 g/g or g/24 h (IQR 4.5)) (Table 2). At the time of the diagnosis, 71.1% of patients (n = 140) had hematuria. Immunosuppressive therapy was used in 53.8% of patients (n = 106). All patients were under renin–angiotensin–aldosterone system (RAAS) inhibitors (Table 1).

At a one-year follow-up, median proteinuria was 0.9 g/g or g/24 h (IQR 2.4). The majority of patients (64.0%, n = 126) had a proteinuria reduction ≥ 25%, and 49.7% (n = 98) had a reduction ≥ 50%. Concerning the level of proteinuria, most patients had proteinuria lower than 1 g/24 h (52.3%, n = 103), 27.4% (n = 54) had 1–3 g/24 h, 9.1% (n = 18) had 3–5 g/24 h and 11.2% (n = 22) had proteinuria ≥ 5 g/24 h. Accordingly, at a one-year follow-up, 27.4% of patients (n = 54) were in complete remission and 48.2% (n = 95) were in partial remission (Table 1). There were no significant differences in proteinuria at the 1-year follow-up between patients with different glomerular diseases (Table 2).

The mean time of observation was 6.5 ± 5.3 years. The eGFR at the end of the follow-up was 72.1 ± 43.3 mL/min/1.73 m^2^, which corresponds to a variation of 9.3 mL/min/1.73 m^2^ (IQR 26.6). The annual decline of the eGFR was 1.4 mL/min/1.73 m^2^ (IQR 5.2). Almost one third of patients had at least a 25% decline in the eGFR (32.5%, n = 64), and 13.2% (n = 26) had a decline ≥ 50%. At the end of the follow-up, 42.6% (n = 84) had an eGFR < 60 mL/min/1.73 m^2^, as shown in Table 1.

### 3.1. Decline of Kidney Function

In this study, 32.5% of patients (n = 64) presented an eGFR reduction ≥ 25% during follow-up. There was no significant difference in age, gender, the presence of hematuria, baseline eGFR or proteinuria (Table 3).

There were no significant differences in eGFR decline between patients with different glomerular diseases (Table 2).

Patients who experienced kidney function decline less frequently had a reduction in proteinuria ≥ 50% (40.6% vs. 54.1%, *p* = 0.076) and less frequently presented a complete remission of proteinuria (21.9% vs. 30.1%, *p* = 0.149) during the first year of follow-up, nearly reaching statistical significance. Concerning the amount of proteinuria, they less frequently had proteinuria < 1 g/24 h (39.1% vs. 58.6%, *p* = 0.010) at the first-year follow-up (Table 3).

At the long-term follow-up, patients who had an eGFR decline ≥ 25% had a significantly lower eGFR (43.4 ± 37.5 mL/min/1.73 m^2^ vs. 85.6 ± 39.1 mL/min/1.73 m^2^, *p* < 0.001) and more frequently had an eGFR < 60 mL/min/1.73 m^2^ (56.3% vs. 30.1%, *p* < 0.001). These patients had a longer follow-up time (7.4 ± 5.3 vs. 6.1 ± 5.2, *p* < 0.001), and the annual decline of the eGFR was significantly higher (5.1 mL/min/1.73 m^2^ (IQR 5.1) vs. 0.4 mL/min/1.73 m^2^ (IQR 3.3), *p* < 0.001) (Table 3).

The reduction in proteinuria to values of < 1 g/24 h, during the first year of diagnosis, was associated with a lesser eGFR decline in the long-term follow-up (log-rank 0.003) (Figure 2).

### 3.2. Predictors of Kidney Function Decline 

We analyzed factors which could predict eGFR decline ≥ 25%. Proteinuria reduction ≥ 50% at the first-year follow-up was associated with lower risk for long-term decline of the eGFR (HR 0.58 (95% CI 0.32–1.06), *p* = 0.077). Proteinuria < 1 g/24 h (HR 0.45 (95% CI 0.25–0.83) *p* = 0.011) at the first-year follow-up was a predictor of long-term decline of the eGFR (Table 4).

A subanalysis was performed to analyze the impact of proteinuria on eGFR decline, independently of the duration of follow-up, and proteinuria < 1 g/24 h maintained a significant association as protective against eGFR decline in the long term (HR 0.30 (95% CI 0.17–0.52) *p* < 0.001).

## 4. Discussion

In this study, we demonstrated that proteinuria reduction during the first-year follow-up was protective against the long-term decline in renal function. A reduction in proteinuria ≥ 50% was associated with a slower decline in renal function. A reduction to levels lower than 1 g/24 h, or reaching complete remission in the first year, was protective against the long-term decline of the GFR.

Several studies showed an association between the degree of proteinuria and a decline in renal function. Berhane et al. studied a population of 2420 diabetic patients and concluded that patients with microalbuminuria at the time of the diagnosis had a risk 2.1 times greater for developing ESKD, and patients with macroalbuminuria were 9.1 times more likely to develop ESKD, when compared with patients without proteinuria [15]. 

A similar association between proteinuria and decline of the kidney function was demonstrated in patients with nondiabetic kidney disease. Ali et al. studied a population of 336 patients with CKD and concluded that patients with a faster decline in the GFR (Δ eTFG ≤ −4 mL/min/1.73 m^2^/year) had proteinuria > 50 g/mol more frequently at the time of the diagnosis (64% vs. 17%, *p* < 0.001) [16]. In fact, most of the studies analyzed the impact of proteinuria at the time of diagnosis on the long-term decline of the GFR.

Reich et al. analyzed 542 patients with primary IgA nephropathy and found that those with sustained proteinuria > 3 g/24 h, during the six-and-a-half-year follow-up, presented a decline in the kidney function 25 times faster than patients with proteinuria < 1 g/24 h (−0.719 ± 0.61 mL/min/1.73 m^2^/month vs. −0.030 ± 0.46 mL/min/1.73 m^2^/month). All patients that reached proteinuria < 1 g/24 h during the follow-up, regardless of the proteinuria concentration at the diagnosis, presented a similar decline in the GFR [17]. As with our study, proteinuria at the time of the diagnosis was not a marker of an accelerated decline of the GFR, in contrast to the proteinuria level at the first-year follow-up, which was independently associated with a greater decline of the GFR.

Proteinuria is a marker of glomerular disease, being the result of filtration through a damaged glomerular basement membrane, but it also has an impact on the decline of the kidney function. It seems that the filtrated proteins exert a direct toxic effect on the tubulointerstitial structure, causing tubular atrophy, inflammation and interstitial fibrosis [12]. Filtrated albumin is responsible for the activation of several proinflammatory cytokines, complement activation and, consequently, tubulointerstitial damage [18,19]. In fact, in patients with diabetic nephropathy, the severity of tubulointerstitial damage is associated with a greater decline of the renal function than in other glomerular diseases [20].

However, we know that the level of proteinuria alone, as a biomarker of CKD, inadequately explains the variability of GFR decline, apart from it being a late marker of GFR decline. Moreover, proteinuria alone does not explain the complex physiopathology of tubular damage [21]. Recent advances in molecular analysis revealed that CKD was closely associated with the dysregulation of numerous metabolites, such as amino acids, lipids, nucleotides and glycoses [22]. Following insult, the kidney tubular cells undergo a cascade of cellular responses that result in the production and accumulation of low-molecular-weight proteins in the urine and systemic circulation that might be exploited as potential biomarkers [23]. Some of these recently identified biomarkers include kidney injury molecule 1 and monocyte chemoattractant protein 1 that represent markers of tubule cell injury, and α_1_-microglobulin and uromodulin that are associated with tubule cell dysfunction [24]. Persistent, low-grade inflammation also seems to have a determinant role in CKD progression [25]. Proinflammatory proteins such as transforming growth factor-β, insulin-like growth factor-1, tumor necrosis factor-α and interleukin-6 were found significantly higher in CKD patients, indicating acute phase response signaling [26]. Therefore, complicated pathomechanisms of CKD development and progression require not a single marker but a combination of markers in order to mirror all the types of alterations occurring in the course of this disease [27,28].

Although these recent studies identified several promising serum and urine biomarkers, which allowed a better understanding of the physiopathology of tubular damage and CKD progression, their cost is still prohibitive, which limits their routine use in clinical practice. Therefore, we continue to rely on classic biomarkers of glomerular and tubular damage, such as proteinuria, which is strongly associated with the decline of kidney function. Therefore, it is essential to outline strategies for the management of patients with proteinuria, in addition to immunosuppressive therapy aimed at the glomerular pathology, and to define the appropriate timing to apply these therapies. Our study showed that proteinuria reduction during the first year after the diagnosis was a predictor of long-term slower decline of the GFR, independently of the therapeutical strategy used for that reduction.

Apart from immunosuppressive therapy used for the treatment of glomerular diseases, there are several antiproteinuric therapies that must be used in order to prevent the decline of the renal function. Renin–angiotensin–aldosterone system (RAAS) inhibitors are the most studied and simultaneously the most effective drugs used for the reduction in proteinuria. By inducing vasodilatation of the efferent arteriola, they reduce intraglomerular pressure, which results in lower proteinuria [29]. In addition, they interfere directly with glomerular basement membrane pores, reducing their permeability to albumin and IgG [30]. In one study of nondiabetic patients, the use of Benazepril, an angiotensin-converting enzyme inhibitor, was associated with a 52% reduction in proteinuria (*p* < 0.001) and a 23% reduction in the rate of decline of the kidney function [31]. According to the REIN study, the use of Ramipril, another ACEI, was associated with a 15% reduction in proteinuria, followed by a significant reduction in the progression to ESKD [32].

Mineralocorticoid receptor antagonists have also been demonstrated to be effective in reducing proteinuria, reducing the inflammatory and profibrotic effect of proteinuria. However, their use is limited by the risk of hypercalemia. A recent meta-analysis that included 1646 patients with CKD showed that adding a mineralocorticoid receptor antagonist to the classic RAAS blockade produced an additional reduction in proteinuria of 38.7%; however patients were three times more likely to develop hypercalemia [33].

More recently, sodium-glucose transport protein 2 (SGLT2) inhibitors have been studied as antiproteinuric drugs. These drugs are currently the first line for the treatment of type 2 diabetes *mellitus*, and have been associated with a proteinuria reduction of 30–50% [34,35,36]. The renoprotective effect of these drugs seems to be the result of hemodynamic changes, in this case, the reduction in intraglomerular pressure, which suggests that they can also be used in nondiabetic nephropathies. A recently published prespecified analysis from the DAPA-CKD trial, which included 4304 patients with diabetic and nondiabetic CKD, revealed that dapagliflozin significantly reduced albuminuria in CKD patients with and without type 2 diabetes, with a larger relative reduction in patients with type 2 diabetes [37].

In our study, 63% of the patients were submitted to immunosuppressive therapy, and 54% presented a proteinuria reduction > 50% at the first-year follow-up. Although not all antiproteinuric drugs were evaluated, all the patients were using RAAS inhibitors, highlighting the importance of these drugs in the treatment of patients with glomerular disease. 

This study has some limitations. First, it is a retrospective study, with a relatively small number of patients, which may affect the extrapolation to the general population. The limited number of patients may also affect the statistical significance of some variables, which may have reached significance with a larger population. Secondly, the method for the quantification of proteinuria was not uniform among the different patients, with some patients having a quantification of proteinuria in a 24 h urine sample and others having a proteinuria to creatinuria ratio based on the first morning urine. We also did not consider the characteristics of kidney biopsies, such as the degree of interstitial fibrosis and tubular atrophy, which may influence the prognosis; instead, we focused on clinical characteristics. However, the analysis of these patients as a whole allowed us to demonstrate the importance of the degree of proteinuria and renal prognosis, regardless of the cause of the glomerular disease. Finally, we did not analyze the immunosuppression used, which may have influenced the rate of reduction in proteinuria and the long-term reduction in the renal function. 

Overall, our study was the first to demonstrate the impact of proteinuria reduction during the first year after diagnosis on the long-term decline of the eGFR, in patients with glomerular disease, regardless of the degree of proteinuria at the time of the diagnosis.

## 5. Conclusions

In this retrospective study of patients with glomerular disease, a proteinuria reduction to lower than 1 g/24 h, during the first year after diagnosis, was a protective factor against the long-term decline of kidney function. Therefore, the first year after the diagnosis of glomerular disease is of utmost importance for the renal prognosis. It is vital to invest in therapeutical strategies to reduce proteinuria in order to delay the decline of the renal function.

## Figures and Tables

**Figure 1 jcm-11-05968-f001:**
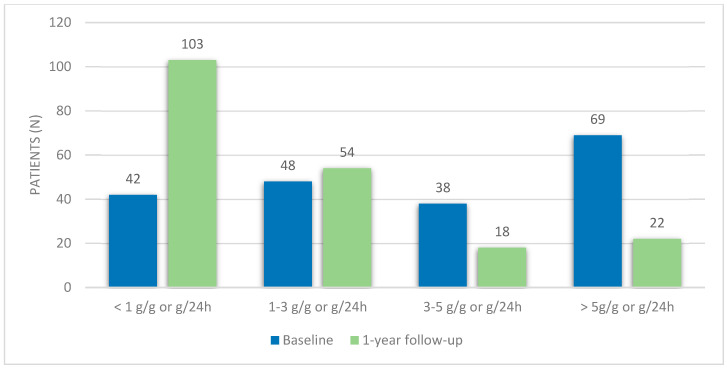
Proteinuria variation during the first-year follow-up.

**Figure 2 jcm-11-05968-f002:**
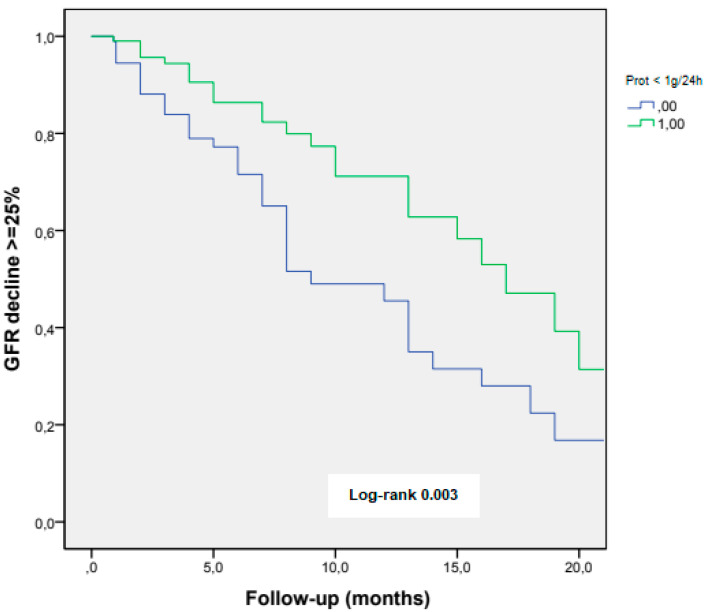
eGFR decline ≥ 25% according to achieving proteinuria < 1 g/24 h at the first-year follow-up.

**Table 1 jcm-11-05968-t001:** Patients’ characteristics.

Characteristics	Patients with Glomerular Disease N = 197
** *Demographics* ** *:*	
Age (years)	41.7 ± 19.7
Gender (male), n (%)	109 (55.3)
*At the diagnosis:*	
Proteinuria	3.5 (IQR 5.8)
eGFR (mL/min/1.73 m^2^)	81.5 ± 49.8
eGFR < 60 mL/min/1.73 m^2^	82 (41.6)
Hematuria, n (%)	140 (71.1)
RAAS inhibitors, n (%)	197 (100)
Immunosuppressive therapy, n (%)	106 (53.8)
*At one-year follow-up:*	
Proteinuria	0.9 (IQR 2.4)
Proteinuria variation (g/g or g/24 h)	1.2 (IQR 4.0)
Proteinuria reduction ≥ 25%, n (%)	126 (64.0)
Proteinuria reduction ≥ 50%, n (%)	98 (49.7)
Complete remission, n (%)	54 (27.4)
Partial remission, n (%)	95 (48.2)
*Long-term follow-up:*	
Time of follow-up (years)	6.5 ± 5.3
eGFR at the end of the follow-up (mL/min/1.73 m^2^)	72.1 ± 43.3
eGFR variation (mL/min/1.73 m^2^)	9.3 (IQR 26.6)
eGFR < 60 mL/min/1.73 m^2^, n (%)	84 (42.6)
eGFR decline, n (%)	134 (68.0)
Annual decline of eGFR (mL/min/1.73 m^2^)	1.4 (IQR 5.2)
Decline of the eGFR ≥ 25%, n (%)	64 (32.5)
Decline of the eGFR ≥ 50%, n (%)	26 (13.2)

**Table 2 jcm-11-05968-t002:** Patients’ characteristics according to glomerular disease.

Glomerular Disease	Minimal Change Disease	Focal Segmental Glomerulosclerosis	Membranoproliferative Glomerulonephritis	IgA Nephropathy	Membranous Nephropathy	*p* Value
**Frequency, n (%)**	36 (18.3%)	63 (31.9%)	21 (10.7%)	56 (28.4%)	21 (10.7%)	
Baseline proteinuria (g/g or g/24 h)	5.8 (IQR 7.2)	4.0 (IQR 5.7)	3.6 (IQR 3.6)	0.9 (IQR 1.4)	6.1 (IQR 4.5)	<0.001
Baseline eGFR (mL/min/1.73 m^2^)	100.9 ± 46.2	75.2 ± 57.0	52.3 ± 20.3	87.8 ± 25.2	79.2 ± 37.3	0.005
1-year proteinuria (g/g or g/24 h)	0.8 (IQR 2.5)	1.4 (IQR 3.0)	1.1 (IQR 1.5)	0.7 (IQR 1.2)	0.9 (IQR 3.9)	0.070
eGFR decline (mL/min/1.73 m^2^)	3.3 (IQR 29.5)	6.3 (IQR 31.0)	2.4 (IQR 41.0)	12.8 (IQR 23.9)	10.5 (IQR 33.0)	0.007
eGFR ≥ 25%, n (%)	5 (13.9%)	25 (39.7%)	8 (38.1%)	20 (35.7%)	6 (28.6%)	0.096

**Table 3 jcm-11-05968-t003:** Patients’ characteristics according to eGFR decline ≥ 25%.

Characteristics	eGFR Decline < 25%, N = 133	eGFR Decline ≥ 25%, N = 64	*p* Value
** *Demographics:* **			
Age (years)	40.2 ± 18.8	45.4 ± 20.7	0.082
Gender (male), n (%)	72 (54.1)	37 (57.8)	0.627
*Baseline Characteristics:*			
eGFR (mL/min/1.73 m^2^)	81.9 ± 44.3	80.7 ± 59.9	0.880
eGFR < 60 mL/min/1.73 m^2^	51 (38.3)	31 (48.4)	0.178
Hematuria, n (%)	92 (69.1)	48 (75.0)	0.398
Proteinuria (g/g or g/24 h)	3.5 (IQR 6.8)	2.8 (IQR 6.4)	0.435
Proteinuria < 1 g/g or g/24 h, n (%)	30 (22.6)	12 (18.8)	0.541
Proteinuria 1–3 g/g or g/24 h, n (%)	26 (19.5)	22 (34.4)	0.023
Proteinuria 3–5 g/g or g/24 h, n (%)	26 (19.5)	12 (18.8)	0.249
Proteinuria > 5 g/g or g/24 h, n (%)	51 (38.3)	18 (28.1)	0.159
Immunosuppressive therapy, n (%)	73 (54.9)	33 (51.5)	0.828
*At one-year follow-up:*			
Proteinuria reduction ≥ 25%, n (%)	88 (66.1)	38 (59.3)	0.353
Proteinuria reduction ≥ 50%, n (%)	72 (54.1)	26 (40.6)	0.076
Proteinuria < 1 g/g or g/24 h, n (%)	78 (58.6)	25 (39.1)	0.010
Proteinuria 1–3 g/g or g/24 h, n (%)	31 (23.3)	23 (35.9)	0.047
Proteinuria < 3 g/g or g/24 h, n (%)	109 (82.0)	48 (75.0)	0.171
Proteinuria 3–5 g/g or g/24 h, n (%)	9 (6.8)	9 (14.1)	0.083
Proteinuria ≥ 5 g/g or g/24 h, n (%)	15 (11.3)	7 (10.9)	0.576
Partial remission, n (%)	69 (51.9)	26 (40.6)	0.092
Complete remission, n (%)	40 (30.1)	14 (21.9)	0.149
*Long-term follow-up:*			
Follow-up time (years)	6.1 ± 5.2	7.4 ± 5.3	0.141
eGFR (mL/min/1.73 m^2^)	85.6 ± 39.1	43.4 ± 37.5	<0.001
eGFR < 60 mL/min/1.73 m^2^, n (%)	40 (30.1)	36 (56.3)	<0.001
eGFR decline (mL/min/1.73 m^2^)	1.8 (IQR 22,7)	31.6 (IQR 28.8)	<0.001
eGFR annual decline (mL/min/1.73 m^2^)	0.4 (IQR 3.3)	5.1 (IQR 5.1)	<0.001

**Table 4 jcm-11-05968-t004:** Predictors of eGFR decline ≥ 25%—univariate analysis.

Characteristics	Hazard Ratio (95% CI)	*p* Value
** *Baseline Characteristics:* **		
Age (years)	1.01 (1.00–1.03)	0.083
Gender (male)	1.16 (0.64–2.12)	0.627
eGFR (mL/min/1.73 m^2^)	1.00 (0.99–1.01)	0.880
eGFR < 60 mL/min/1.73 m^2^	1.51 (0.83–2.76)	0.179
Hematuria	1.34 (0.68–2.63)	0.339
Proteinuria (g/g or g/24 h)	0.97 (0.91–1.04)	0.434
Proteinuria < 1 g/g or g/24 h Proteinuria 1–3 g/g or g/24 h Proteinuria 3–5 g/g or g/24 h Proteinuria ≥ 5 g/g or g/24 h	0.79 (0.38–1.67) 2.16 (1.10–4.22) 0.70 (0.39–1.28) 0.63 (0.33–1.20)	0.542 0.025 0.249 0.161
Immunosuppressive therapy	0.94 (0.51–1.71)	0.828
*At one-year follow-up:*		
Proteinuria reduction ≥ 25%	0.75 (0.40–1.38)	0.353
Proteinuria reduction ≥ 50%	0.58 (0.32–1.06)	0.077
Proteinuria < 1 g/g or g/24 h	0.45 (0.25–0.83)	0.011
Partial remission	0.64 (0.35–1.16)	0.140
Complete remission	1.22 (0.32–1.31)	0.229

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
