# Peer review of "Impact of Early Proteinuria Reduction in Glomerular Disease and Decline of Kidney Function: A Retrospective Cohort"

_jcm, 2022, doi:10.3390/jcm11195968_

Round 1
Reviewer 1 Report
Marques et al studied the association of proteinuria with outcome in patients with glomerular disease in a single center in Lisbon, Portugal. Their study, with a relevant number of included patients, is of interest but not novel. The manuscript is comprehensible but contains some syntax and grammar errors. It would definitely be strengthened by including diagnoses and histopathological findings. The manuscript can be improved by changing the tables and figures according to my comments below.
Major
- Please indicate this is a single center study at the end of the introduction.
- Part 2.1: why would you exclude other glomerular diseases such as pauci-immune glomerulonephritis or amyloidosis? What was the rationale behind your inclusion criteria?
- Why would you exclude patients with eGFR<15? Please motivate.
- Which patients were excluded? How was the screening procedure performed?
- Were diagnoses all confirmed by kidney biopsy? If so, are histological parameters available?
- In the methods it is mentioned that data were collected in 2019, but if mean observation time is 6.6 years, this is impossible, you must have collected data beyond 2019?
- Line 98: eGFR has a standard deviation of 47.3. Was this variable normally distributed? Line 112 same question.
- Line 99: the standard deviation of proteinuria is larger than the mean which certainly indicates it is not normally distributed. In that case, the variable should be presented as median with IQR or range.
- Did any patients already have SGLT2 inhibitors?
- Figure 3: please review your figures, especially substituting the Portuguese with English. (Moito obrigado)
- Please provide figures of eGFR decline (numerically) and proteinuria follow-up without caterogical divisions (such as current fig 1-3) but with stratification according to histopathological diagnosis.
- Discussion line 213: I don’t understand how this is relevant in this manuscript.
- Line 240: please correct this sentence.
- Line 248: please temper or adjust this statement, the fact that proteinuria results in worse kidney outcome has been known for many years, also in non-diabetic kidney disease. In other words, you were not the first. E.g., PMID 33514642, PMID 33555575, PMID 34670811, PMID 33615070, etc.
- Is it possible to add diagnosis to the variables? This is quite essential and would seriously strengthen your study.
- I would remove the categorical distribution of proteinuria from the table 1 and show this in a figure, at baseline and at follow-up (in one figure); same for eGFR
Minor
- Line 27: “e” should be “and”
- Line 41: “Proteinuria […] associated [+with a] greater risk”
- Line 231: please correct RASS
Round 2
Reviewer 2 Report
The authors modified the paper, but not sufficiently. Adding references without substantially modifying the discussion does not complete the comments to the paper.
Unfortunately, the changes are insufficient
